# Comparison of the spatial and temporal distribution of cutaneous and mucosal leishmaniasis in the state of Rio de Janeiro between 2001 and 2011

**Lucia Regina do Nascimento Brahim Paes**[1], **Raquel de Vasconcellos Carvalhaes de Oliveira**[1], **Monica de Avelar F. M. Magalhães**[2], **Maria Inês Fernandes Pimentel**[1], **Marcelo Rosandiski Lyra**[1], **Luiz Eduardo Carvalho-Paes**[3], **Ananda Dutra da Costa**[1], **Cristina Maria Giodarno Dias**[4], **Anísia Darc do Nascimento Brahim**[5], **Bruno Moreira de Carvalho**[3], **Claudia Cristina Jardim Duarte**[1], **Mauro Celio de Almeida Marzochi**[1], **Ester Cleisla dos Anjos Soares**[1], **Armando de Oliveira Schubach**[1,6], **Cláudia Maria Valete-Rosalino**[1,6,7] *

1 Evandro Chagas National Institute of Infectious Disease, Oswaldo Cruz Foundation, Rio de Janeiro, RJ, Brazil, 2 Institute of Scientific and Technological Communication and Information, Oswaldo Cruz Foundation, Rio de Janeiro, RJ, Brazil, 3 Oswaldo Cruz Institute, Oswaldo Cruz Foundation, Rio de Janeiro, RJ, Brazil, 4 State Department of Health of Rio de Janeiro, Epidemiological Surveillance, Rio de Janeiro, RJ, Brazil, 5 Deputy Director of Labor Management, Oswaldo Cruz Foundation, Rio de Janeiro, RJ, Brazil, 6 Productivity Fellowship Holder of the Brazilian National Council for Scientific and Technological Development (CNPQ), Brasília, Brazil, 7 Department of Otorhinolaryngology and Ophtalmology, Federal University of Rio de Janeiro, RJ, Brazil

* claudia.valete@ini.fiocruz.br, cmvalete@gmail.com

## Abstract

### Objective

To compare the spatio-temporal distribution of cutaneous leishmaniasis (CL) cases with mucosal leishmaniasis (ML) cases in the state of Rio de Janeiro (RJ) between 2001 and 2011.

### Method

The incidence rates (IR) of CL and ML were calculated for the cases notified between 2001–2011 in the Information System of Notifiable Diseases for Rio de Janeiro (RJ, and for the municipalities of Rio de Janeiro and Angra dos Reis, with georeferencing and construction of thematic maps. A negative binomial regression model was used to assess the temporal dependency between CL and ML.

### Results

Higher IR of CL and ML were observed up to 2006. The cases of CL and ML increased annually concomitantly in the state of RJ and in Angra dos Reis, even when they were controlled by the CL rates of the previous year. The municipality of Rio de Janeiro presented smaller annual CL IR after the occurrence of high ML IR in the two previous years.

**Data Availability Statement:** All relevant data are within the manuscript and its Supporting Information files.

**Funding:** This study was financed in part by the Coordenação de Aperfeiçoamento de Pessoal de Nível Superior - Brasil (CAPES) - Finance Code 001.

**Competing interests:** The authors have declared that no competing interests exist.

## Conclusion

The temporal association observed between CL and ML suggests that: either the mucosal lesions were already incipient from the beginning of CL manifestation, or the *Leishmania* species circulating in RJ is able to produce early mucosal lesions.

## Introduction

American tegumentary Leishmaniasis (ATL) represents a significant health challenge in four global ecoepidemiological regions: the Americas, East Africa, North Africa, and West and Southeast Asia [1]. Between 2017 and 2022, the Pan American Health Organization (PAHO) reported a total of 252,998 cases of cutaneous leishmaniasis (CL) and mucosal leishmaniasis (ML), with an annual average of 42,166 cases. Of these, approximately 97% were attributed to the following sub-regions: Andean Area (41%), Brazil (37%), and Central America (19%) [2]. In the Brazilian context, in 2021, there were 15,023 cases of leishmaniasis registered, covering all regions of the country, with 14,260 (95%) cases being of the cutaneous leishmaniasis and 756 (5%) of the mucosal leishmaniasis. However, in 2022, a reduction in the number of reported cases was observed, totaling 12,878 [1]. *Leishmania (Viannia) braziliensis* is the species that more frequently causes ATL in Brazil. In the state of Rio de Janeiro, ATL is almost exclusively caused by this *Leishmania* species and it occurs in areas where phlebotomine vectors of peridomestic habits, such as *Lutzomyia intermedia* and *Lutzomyia migonei*, predominate [3–6].

Frequently, the infection occurs in domestic dogs, horses and eventually in domestic cats [7–9]. To date, there is no conclusive information on the involvement of wild and synanthropic animals with *L. (V.) braziliensis* in the state of Rio de Janeiro [10]. *L. (V.) braziliensis* could therefore be being introduced in modified environments located in new areas of the Brazilian Southeast Region, through infected humans or their domestic animals. They would act as infection sources for sand flies adapted to the environment around the houses [10,11]. Human secondary cases of cutaneous leishmaniasis (CL) would happen and consequently mucosal leishmaniasis (ML) cases would occur. The last ones would happen in the inverse ratio to the degree of endemicity [11].

To our knowledge, only one autochthonous case of L. (Leishmania) amazonensis in 2015 and two uncommon cases of cutaneous leishmaniasis (CL) caused by L. (L.) infantum have been described in the state of Rio de Janeiro in 2007 [12,13]. The incubation period of the disease in humans generally ranges from two weeks to two months and the patient, even when treated, can house *L. (V.) braziliensis*, which may be viable in skin cultures from lesion scars up to 12 years after clinical healing [3,14]. This could turn the human being into a potential source of infection for the vector. The long and variable period of being infected after treatment could also hamper the epidemiological analyses of correlation between the occurrence or the introduction of new cases and the possible outbreak of secondary cases of CL and ML.

Cutaneous leishmaniasis is the most common manifestation of ATL, presenting exclusively cutaneous lesions, which tend to heal, according to the host's cellular immunity response. They are generally solitary or few in number, rounded or oval, with a coarse-grained bottom, well-defined raised edges and an infiltrated, erythematous base with a firm consistency.

On the other hand, mucosal leishmaniasis, whose main etiological agent is L. (V.) braziliensis, being a more serious form, and may manifest itself concomitantly with the initial skin lesion or weeks, years or even decades after its healing. It compromises the mucous

membranes of the upper airways and digestive tract. Its pathogenesis is not yet fully under-stood. The nasal mucosa is where most of the lesions are found, alone or associated with other locations, is involved in 91.3% of LM cases, followed by the mouth (37.9%), pharynx (31.4%) and larynx (30.1%).pacient [3,15,16].

It has been reported that in 50% of the cases of CL patients that evolve into ML this occurs in the first two years after cutaneous lesion healing, while in 90% it occurs within 10 years [7–9]. In Brazil, ML cases represent 3 to 6% of notified ATL cases, although in some municipali-ties the number exceeds 25% [3,11,17]. The hypothesis that a better parasite-host adaptation can be responsible for the lower number of ML cases in areas of older endemics, such as the Brazilian North and Northeast Regions, has been suggested [3]. The Laboratory of Clinical Research and Surveillance in Leishmaniasis of the Evandro Chagas National Institute of Infec-tious Diseases-Fiocruz (LaPClinVigiLeish/INI/FIOCRUZ), in the municipality of Rio de Janeiro, is responsible for the treatment of most ML patients notified in the–Rio de Janeiro state [18]. Studies in that institution have revealed that, for in that State, the concomitant mucosal leishmaniasis (simultaneous presence of cutaneous and mucosal lesions) is the most common manifestation of ML (37.9% cases), followed by ML of undetermined origin (with no previous history of CL or without suggestive cutaneous scars) with 26.7% cases and, in the third place, the late mucosal leishmaniasis (with previous history of CL or suggestive cutaneous scars) with 25.2% cases [15]. The latter is considered the most frequent manifestation in several regions of Brazil [19].

In Brazil, an annual mean of 25,763 new ATL cases was reported from 1995 to 2014, with an average detection coefficient of 14.7 cases per 100.000 inhabitants. Particularly, in the state of Rio de Janeiro, the highest incidence rate of ATL was observed in 2005, with 351 cases reported. Cutaneous leishmaniasis (CL) prevailed with 87.90% cases, followed by mucosal leishmaniasis (ML) cases (11.83%) and non-classified forms (0.27%). During that period (2004–2013), there was a reduction of 86.46% in CL incidence and of 75% in ML cases in the state of Rio de Janeiro. Although we carried out the analyzes in 92 municipalities in the State of Rio de Janeiro, the municipalities of Angra dos Reis and Rio de Janeiro (capital of the State) were chosen for this study, as they were the ones that presented the most significant data in relation to LC and LM incidence rates from 2001 to 2011 [11].

The analysis of ATL spatial and temporal distribution could help to understand the dynam-ics of the disease. However, there are few studies with this analysis in Brazil and particularly in Rio de Janeiro [17,19,20]. This study aimed to compare the distribution of CL and ML cases in Rio de Janeiro state, as well as in the municipalities of Rio de Janeiro and Angra dos Reis, from 2001 to 2011.

## Materials and methods

### Study area

The state of Rio de Janeiro is one of the 27 federative units in Brazil. It is located in the eastern portion of the Southeast region and occupies an area of 43.696.054 km$^2$. The state of Rio de Janeiro is part of the biome of the Brazilian Atlantic Forest, with mountains and lowlands located between the mountainous Serra da Mantiqueira and the Atlantic Ocean, composing diversified landscapes. The state is formed by two morphologically distinct regions, the low-land and a plateau, and has a tropical climate [21]. The municipality of Rio de Janeiro is the capital of the state and the second largest conurbation in Brazil. It occupies an area of 1.200.179 Km$^2$, and is divided into 160 neighborhoods, grouped into 33 administrative regions and seven district councils. The western region concentrates the most populous neighbor-hoods in the municipality, and it has a high population growth, although not all with a similar

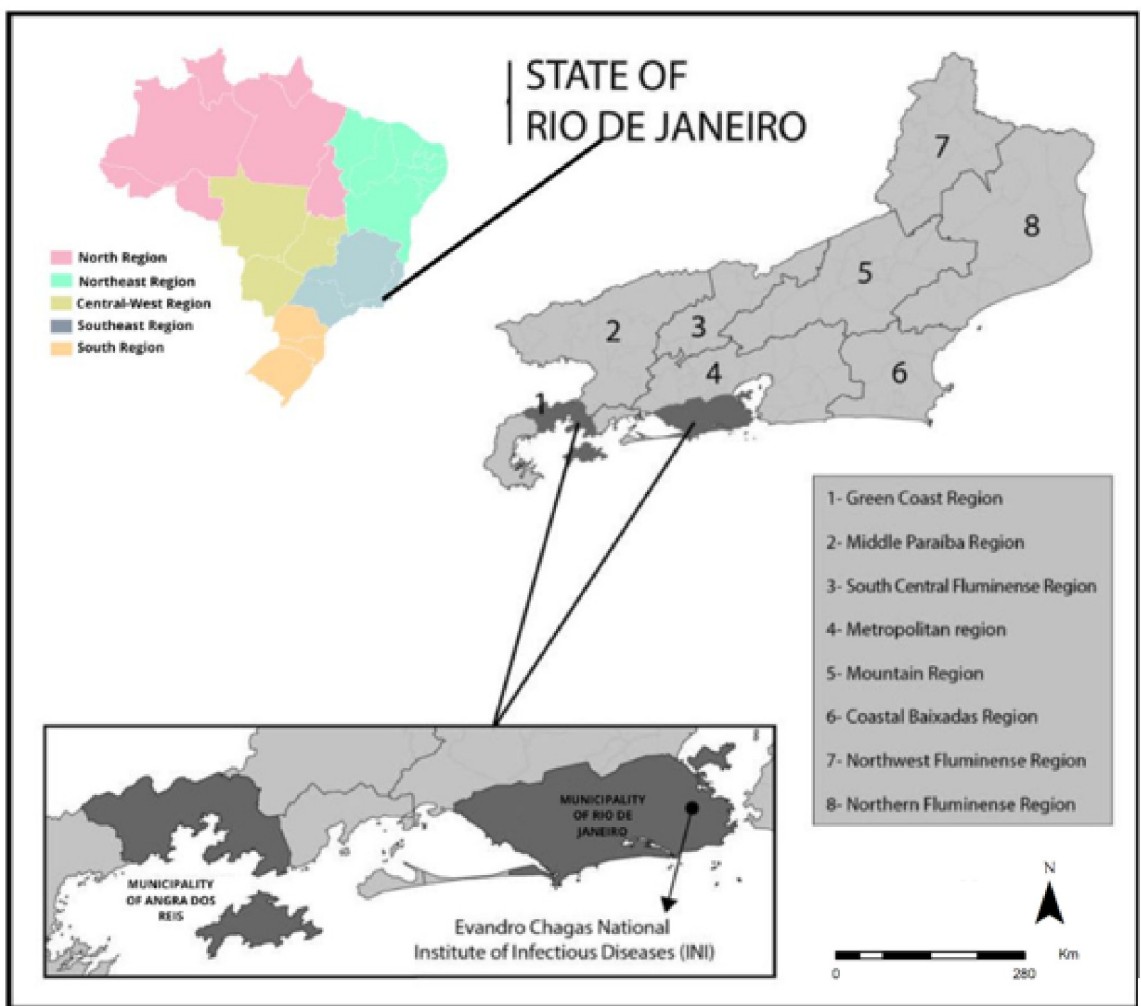

**Fig 1. Map of Brazil, evidencing the State of Rio de Janeiro and its Region, the municipalities of Angra dos Reis and Rio de Janeiro and the localization of the Evandro Chagas National Institute of Infectious Diseases- Oswaldo Cruz Foundation.** Figure created by the authors Fiocruz 2017.

development, which causes inappropriate agglomerations and segregations. Rio de Janeiro municipality has a tropical Atlantic climate, and the average annual temperature is 23.8˚C. Since it is a coastal city, the effect of the proximity to the sea is noticeable, resulting in relatively small thermal amplitudes. The annual average of maximum monthly temperatures is 27.3˚C, and of minimum monthly temperatures is 21˚C [22] (Fig 1).

The municipality of Angra dos Reis is located in the south of the state of Rio de Janeiro, in the Green Coast Region, 158 km away from the state capital. It has an area of 825.082 Km$^2$, at an average altitude of six meters, and comprises 365 islands (IBGE, 2016). Its current importance comes from the Angra dos Reis port, which is one of the busiest in the country and also because the only nuclear power plants in operation in Brazil are in the municipality. It is also one of the highest known tourist places of the state and of the country. Most of the city is surrounded by hills, and about 36% of the population lives in slums, located in the hills or mangrove areas, which places the municipality in the tenth place among the Brazilian cities, regarding the proportion of households in slums in the country [23]. According to data from the Brazilian National Institute of Meteorology, the minimum absolute temperature registered

in Angra dos Reis, since 1961, was 9.4°C on August 12, 1988, and the maximum absolute temperature was 39.3°C on February, 12, 1966. The mean of the maximum temperatures was 27°C and the mean of the minimum temperatures was 19.4°C for the same period [22] (Fig 1).

## Population and ATL notification data

This is an ecologic study, based on the notification data of ATL cases in the state of Rio de Janeiro, from the Information System of Notifiable Diseases (Sistema de Informação de Agravos de Notificação-SINAN), obtained at the Rio de Janeiro State Health Department (SESRJ), considering municipal population data obtained in the 2001 and 2011 censuses and the intercensity projections from 2002 to 2010 of the Brazilian Institute of Geography and Statistics [22]. This data allowed to calculate the incidence rate of CL by 100.000 inhabitants (number of new CL cases in the year/population of the municipality x 100.000) and of ML (number of new ML cases in the year/population in the municipality x 100.000).

## Map of CL and ML occurrence in the municipalities of the state of Rio de Janeiro (2001–2011)

A Geographic Information System (GIS) was used to assess the spatiotemporal distribution of the disease cases, which were georeferenced by municipality. Thematic maps of disease rates per year were created, helping to detect TL dispersion patterns in the state of Rio de Janeiro and in the spatial and temporal analysis between the occurrence of LM and LC. The ArcGis 10.4 program was used to analyze the results and prepare the maps referring to Figs 2 and 3. The program was made available by the Geoprocessing Center/ Institute of Communication and Scientific and Technological Information in Health.

## Variables

The minimum, maximum, and median of the time elapsed between the observation of CL cases and ML cases were obtained for Rio de Janeiro state from 2001 to 2011. For each year, CL and ML incidence rates were obtained by the ratio between the cases for each clinical form and the population count, expressed per 100,000 inhabitants.

An exploratory analysis of the annual CL and ML rates time-series by line charts was performed to describe the temporal evolution of the CL and ML incidence rates. The temporal analysis was carried out for the entire state of Rio de Janeiro (92 municipalities) and since the municipalities of Rio de Janeiro and Angra dos Reis had the highest rates, they were selected for the study. Due to the number of cases in the other municipalities, it was not possible to see different patterns in each of the other 90 municipalities.

## Statistical analysis

Negative binomial regression models were used to evaluate the effects of the incidence rate annual series of CL and ML (2001–2011). This modeling approach was performed separately for each scenario: Rio de Janeiro state, Angra dos Reis, and Rio de Janeiro city-data. Considering the small number of ML cases, we used the count of CL cases in the year $t$ as an outcome variable with offset *(log (population in the year t))* to model the CL incidence rates, for example: *log (CL cases $_t$) = $\beta_0 + \beta_1 X_{1t} + log (pop_t)$* where $t$ is a year t, $X_1$ is an explanatory variable and *pop* is the population count. The four regression models were adjusted using ML and CL rates (variables $X_1$ and $X_2$, respectively) lagged in previous years as explanatory variables: model A– ML rate of the previous year (*log (CL cases $_t$) = $\beta_0 + \beta_1 X_{1(t-1)} + log (pop_t)$*, where $X_{1(t-1)}$ is the ML rate of the year *t-1*); model B–ML rate of the two previous years (*log (CL cases $_t$) = $\beta_0 + \beta_1 X_{1(t-}$*

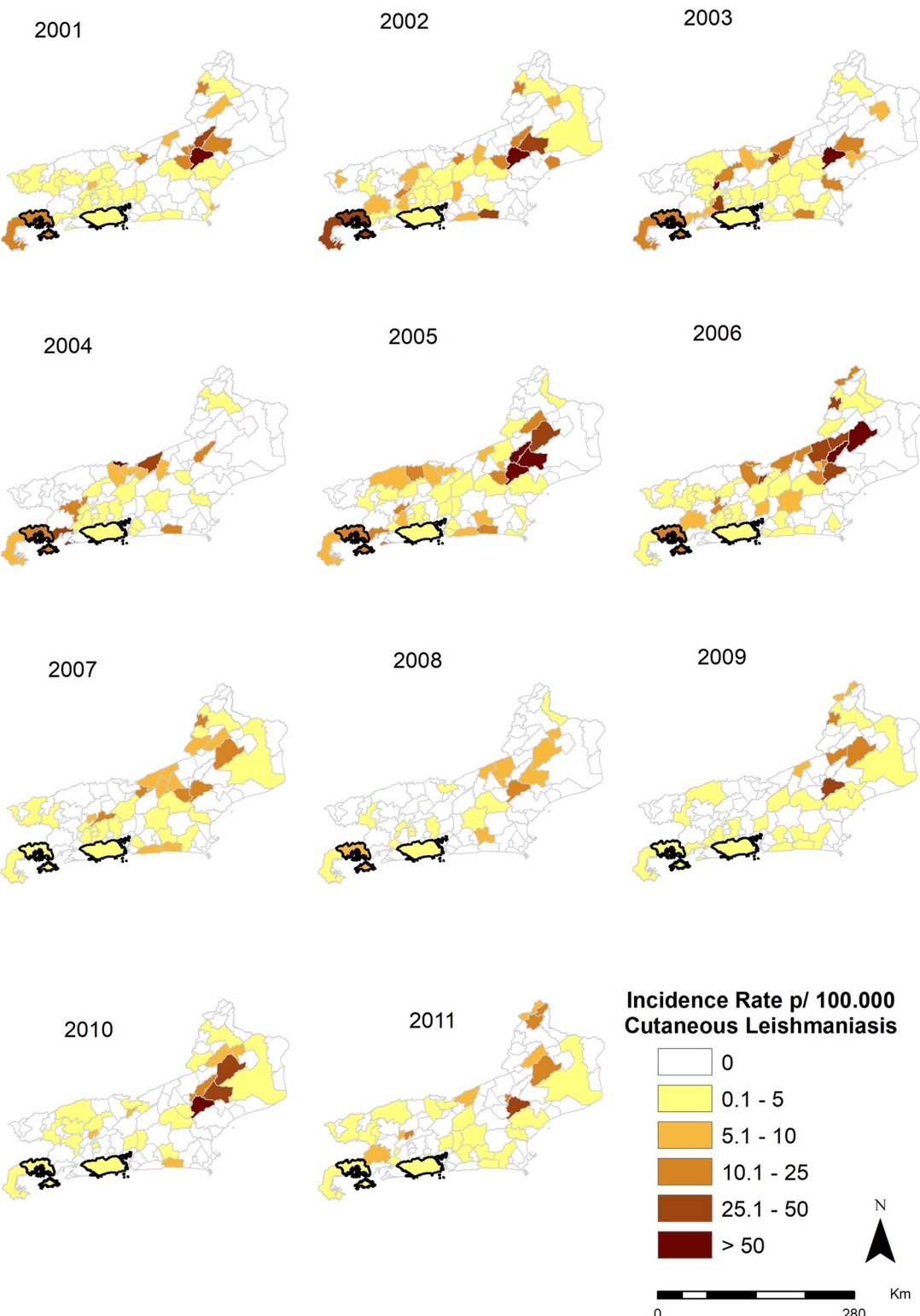

**Fig 2. Distribution of the incidence of cutaneous leishmaniasis in the state of Rio de Janeiro, Brazil, from 2001 to 2011 with emphasis on the municipalities of interest in the study of Rio de Janeiro and Angra dos Reis.** Figure created by the authors Fiocruz 2017.

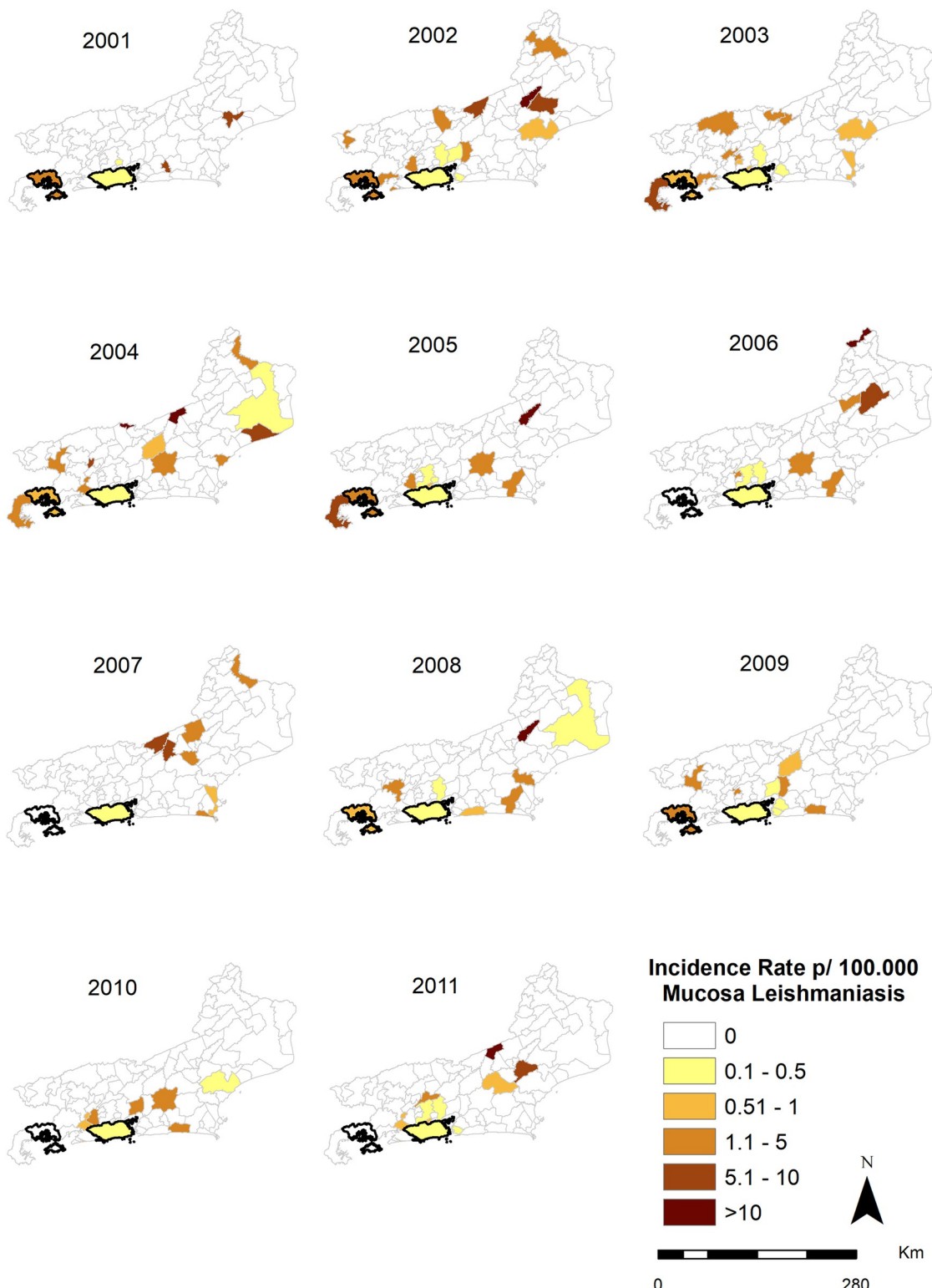

**Fig 3. Distribution of the incidence of mucosal leishmaniasis in the state of Rio de Janeiro, Brazil, from 2001 to 2011, highlighting the municipalities of interest in the study of Rio de Janeiro and Angra dos Reis.** Figure created by the authors Fiocruz, 2017.

$_{2)}$+ $log$ ($pop$ $_t$), where $X_{1(t-2)}$ is the ML rate of the year $t$-2); model C- ML rate for the year of study controlled by the CL rate of the previous year ($log$ ($CL$ $cases$ $_t$) = $\beta_0$+ $\beta_1 X_{1t}$+$\beta_2 X_{2(t-1)}$+$log$ ($pop$ $_t$), where $X_{1t}$ is the ML rate of the year $t$ e $X_{2(t-1)}$ is the CL rate of the year $t$-1) and model D–ML rate of the two previous years controlled by CL rate of the previous year ($log(CL$ $cases$ $_t)$ = $\beta_0$+$\beta_1 X_{1(t-2)}$+$\beta_2 X_{2(t-1)}$+$log$ ($pop$ $_t$), where $X_{1(t-2)}$ is the ML rate of the year $t$-2 e $X_{2(t-1)}$ is the CL rate of the year $t$-1).

For each scenario (Rio de Janeiro state, Angra dos Reis municipality, and Rio de Janeiro municipality), one of these four adjusted models was considered the best model: when it had $\beta_1$ (ML rate effect) significant in the Wald test and a smaller Akaike information criterion (AIC). The $\beta_2$ effect was considered in the model as a potential controlled factor, so it was not a criterium to choose the final model. Additionally, the goodness-of-fit of the final model was assessed by the residual deviance function.

The model was interpreted by the coefficients $\beta_1$ or $\beta_2$ and by CL predicted values of the best-adjusted model.

P-values<0.05 indicated significant values in the statistical tests.

The analyses were performed in the MASS library of the R software version 3.2.3.

## Ethics statement

This Project was approved by the Ethics in Research Committee of Evandro Chagas National Institute of Infectious Diseases (CAAE 17222113.2.0000.5262).

## Results

The CL and ML incidence rates from 2001 to 2011 are presented in thematic maps (Figs 2 and 3). It is possible to observe that ML cases occurred in the same year or up to 5years after the CL cases, with median of 1 year between both clinical forms. The municipality of Rio de Janeiro was the only municipality in the state that had CL and ML cases throughout all years of the study period. The Green Coast Region was the only one in the state of Rio de Janeiro with CL and ML cases in all its municipalities, including Angra dos Reis.

Models A and B did not present a p value of β2 for the State of Rio de Janeiro and the municipalities of Angra dos Reis and Rio de Janeiro. Models C and D present the highest rates, with model C having a significant p-value (Table 1).

### • State of Rio de Janeiro

Fig 4 shows the times-series distribution of ATL clinical forms in the Rio de Janeiro state, from 2001 to 2011. There is an increased incidence of CL from 2001 to 2006, with peaks in 2002, 2005 and 2006, and increase in ML incidence from 2002 to 2006 and from 2008 to 2009.

The negative binomial regression models A, B, and D did not present significant β1 coefficients (ML rate effect) for the explanation of CL incidence rate. Model C showed that ML incidence rate (a positive β1) tends to increase concomitantly with CL annual incidence rate, even when controlled by the CL rate of the previous year (β2). This model C had a good fit according to the deviance function (p = 0.1831) and an AIC = 112.45.

The CL incidence rate values predicted by 100,000 inhabitants by model C are shown in Fig 5.

### • Municipality of Rio de Janeiro

Fig 6 shows the temporal evolution of ATL clinical manifestation forms in the municipality of Rio de Janeiro, from 2001 to 2011. It is possible to observe an increase in annual CL

**Table 1. Model A: log (CL cases t) = β0+β1X1(t-1)+log (popt), where X1(t-1) is the ML rate of the year t-1.** Model B: log (CL cases t) = β0+β1X1(t-2)+ log (pop t), where X1(t-2) is the ML rate of the year t-2. Model C: log (CL cases t) = β0+ β1X1t+β2X2(t-1)+log (pop t), where X1t is the ML rate of the year t. Model D: log(CL cases t) = β0+β1X1(t-2)+β2X2(t-1)+log (pop t), where X1(t-2) is the ML rate of the year t-2 e X2(t-1) is the CL rate of the year t-1.

| | Rio de Janeiro state | | | |
| --- | --- | --- | --- | --- |
| | **Model A** | **Model B** | **Model C** | **Model D** |
| $\beta_1$ (p-value) | 6.4519 (0.108) | 2.9828 (0.518) | **10.0610 (<0.001)** | -2.4487 (0.523) |
| $\beta_2$ (p-value) | | | 0.5020 (0.004) | 0.8322 (0.001) |
| AIC | 122.11 | 111.18 | 112.45 | 106.4 |
| | Rio de Janeiro city | | | |
| | **Model A** | **Model B** | **Model C** | **Model D** |
| $\beta_1$ (p-value) | xxx | (0.685) | **6.9927 (0.0231)** | **-8.3269 (0.032)** |
| $\beta_2$ (p-value) | | | 0.3052 (0.3039) | 1.139 (0.001) |
| AIC | 98.75 | 92.223 | 97.56 | 87.58 |
| | Angra dos Reis city | | | |
| | **Model A** | **Model B** | **Model C** | **Model D** |
| $\beta_1$ (p-value) | **0.7111 (0.015)** | 0.3248 (0.364) | **11.8321 (<0.001)** | -0.6552 (0.087) |
| $\beta_2$ (p-value) | | | 0.07424 (0.002) | 0.10455 (0.001) |
| AIC | 79.516 | 71.735 | 79.525 | 67.526 |

incidence in 2002 and from 2004 to 2006, while the ML incidence increase in 2002, from 2004 to 2006, in 2008, and in 2011.

The negative binomial regression model B did not present significant $\beta_1$ coefficient in the explanation of CL incidence rate. The models C and D were considered the best models, due to their $\beta_1$ significants and their low AIC values, when compared to model A.

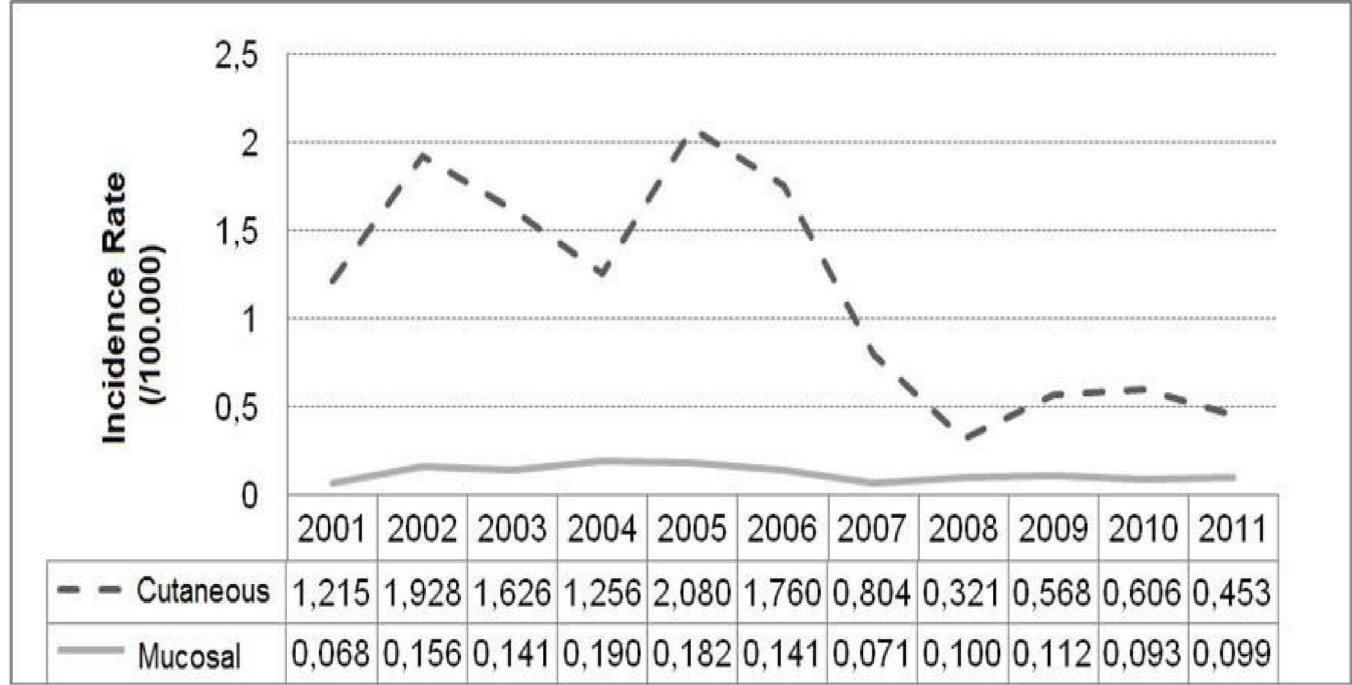

**Fig 4. Temporal evolution of the incidence rates according to tegumentary leishmaniasis clinical manifestations in the state of Rio de Janeiro, Brazil, from 2001 to 2011.** FIOCRUZ, 2017.

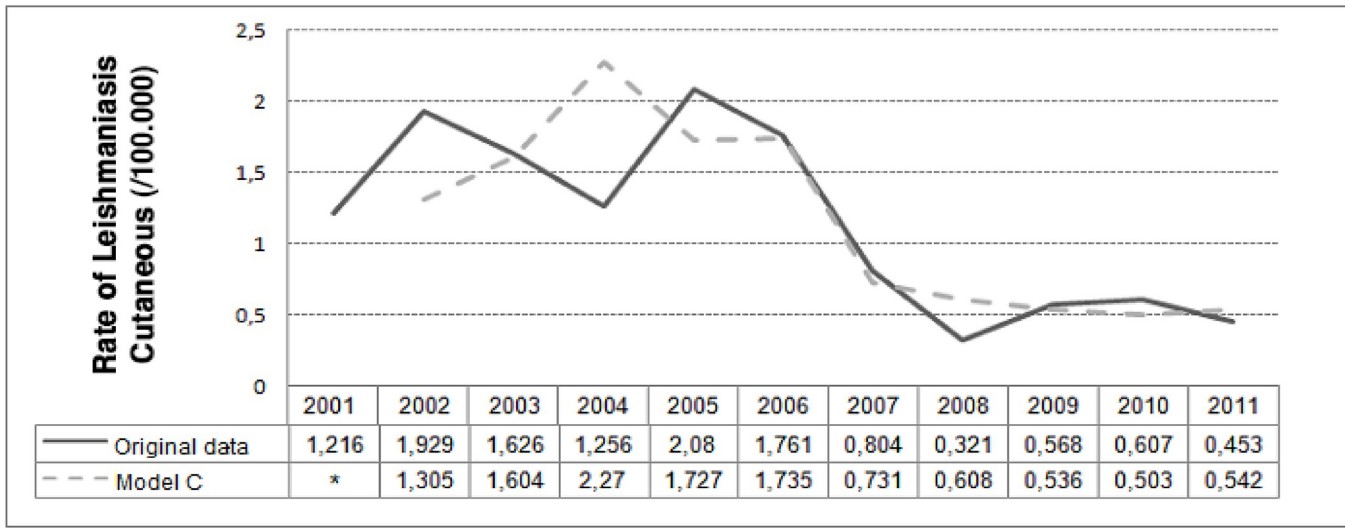

**Fig 5. Prediction of cutaneous leishmaniasis rates (/100,000) in the state of Rio de Janeiro, Brazil, by the negative binomial models, from 2001 to 2011.** Model C: Cutaneous rate = Mucosal rate+ Cutaneous rate of the previous year. * Predicted values of rates not calculated due to the dependency of mucosal leishmaniasis incidence rate of the previous years FIOCRUZ, 2017.

Model C showed that ML rate (a positive β1) tends to increase concomitantly with CL rate of the same year, even when controlled by the CL rate of the previous year (β2). Model D showed that the highest the rate of ML of the two previous years (a negative $\beta_1$) the lowest the

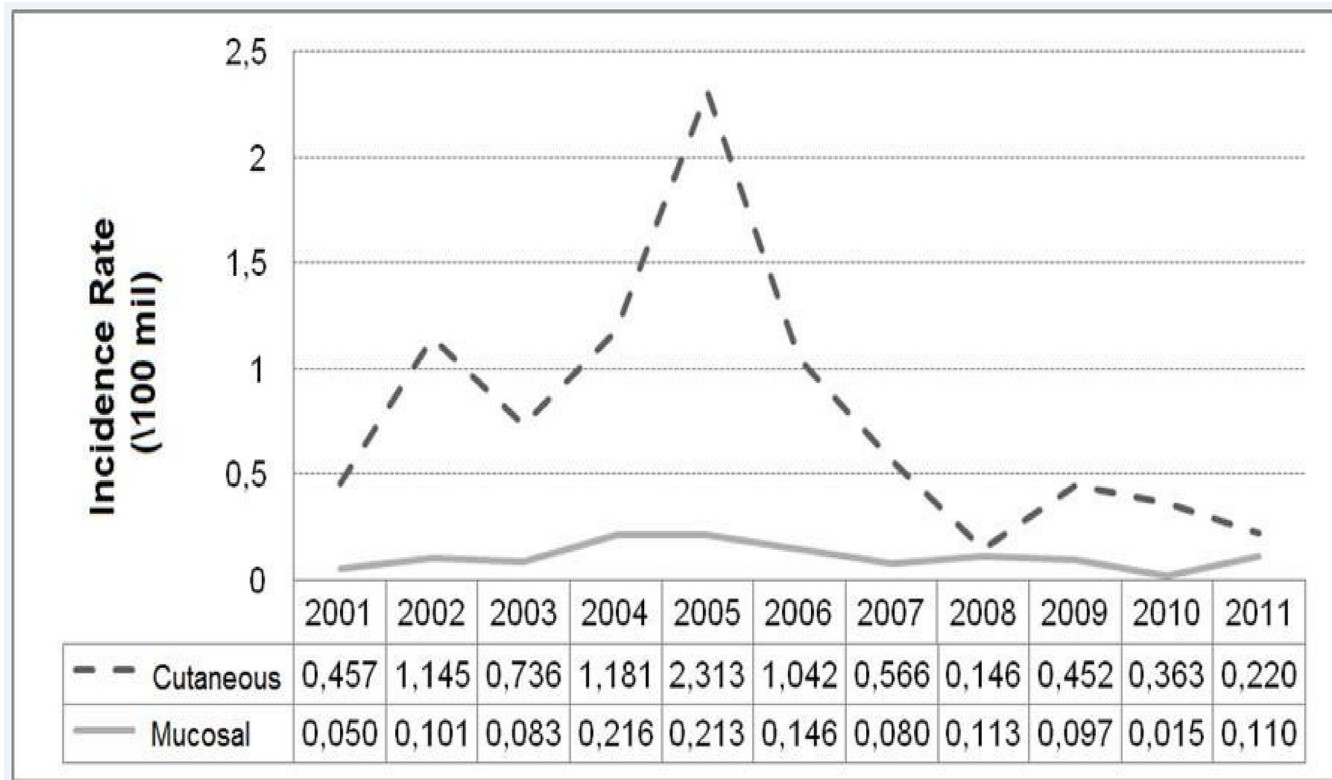

**Fig 6. Temporal evolution of the incidence rates according to tegumentary leishmaniasis clinical manifestations in the municipality of Rio de Janeiro, Brazil, from 2001 to 2011.** FIOCRUZ, 2017.

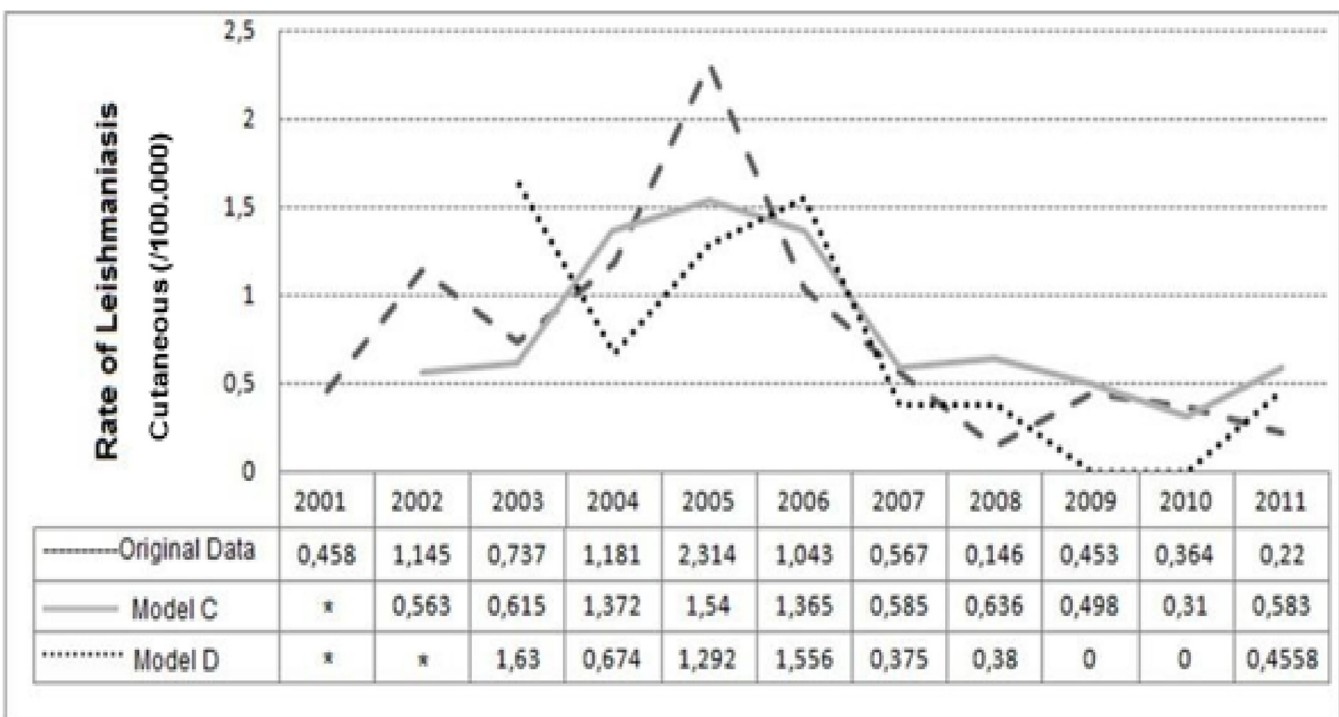

| | 2001 | 2002 | 2003 | 2004 | 2005 | 2006 | 2007 | 2008 | 2009 | 2010 | 2011 |
|---|---|---|---|---|---|---|---|---|---|---|---|
| --------Original Data | 0,458 | 1,145 | 0,737 | 1,181 | 2,314 | 1,043 | 0,567 | 0,146 | 0,453 | 0,364 | 0,22 |
| ——— Model C | * | 0,563 | 0,615 | 1,372 | 1,54 | 1,365 | 0,585 | 0,636 | 0,498 | 0,31 | 0,583 |
| ·········· Model D | * | * | 1,63 | 0,674 | 1,292 | 1,556 | 0,375 | 0,38 | 0 | 0 | 0,4558 |

**Fig 7. Prediction of cutaneous leishmaniasis rate(/100,000) in the municipality of Rio de Janeiro, Brazil, by the negative binomial models, from 2001 to 2011.** Model C: Cutaneous rate = Mucosal rate + Cutaneous rate of the previous year. Model D: Cutaneous rate = Mucosal rate of the 2 previous years + Cutaneous rate of the previous year. * Predicted values of rates not calculated due to the dependency of mucosal leishmaniasis incidence rate of the previous years.

incidence of CL in the year of study, even when controlled by CL rate of the previous year ($\beta_2$). Models C and D had good fit according to the deviance function (p = 0.1609, p = 0.1597, respectively). However, model D was considered the one with the best fit, because of the lowest AIC value.

The values predicted for CL incidence rate by 100.000 inhabitants estimated by models C and D are shown in Fig 7.

### • Municipality of Angra dos Reis

Fig 8 shows the temporal evolution of ATL clinical manifestation forms in the municipality of Angra dos Reis, from 2001 to 2011. Greater incidence of CL between 2001 and 2006 is observed, with incidence reduction from 2007 on, while ML incidence had periods of greater values in 2001, 2002, 2005 and 2009, with periods with no ML cases between 2006–2007 and 2010–2011.

The negative regression models B and D did not present significant coefficients in the out-comeCL incidence rate.

Model A showed that the highest the ML rate in the previous year (a positive $\beta_1$), the highest the CL in the study year. Model C showed that ML rate (a positive $\beta_1$, p<0.001) tends to increase concomitantly to CL rate in the study year, even when controlled by CL rate of the previous year ($\beta_2$).

Models A and C had good fit according to the deviance function (p = 0.2091 and p = 0.1522, respectively) and with similar AIC.

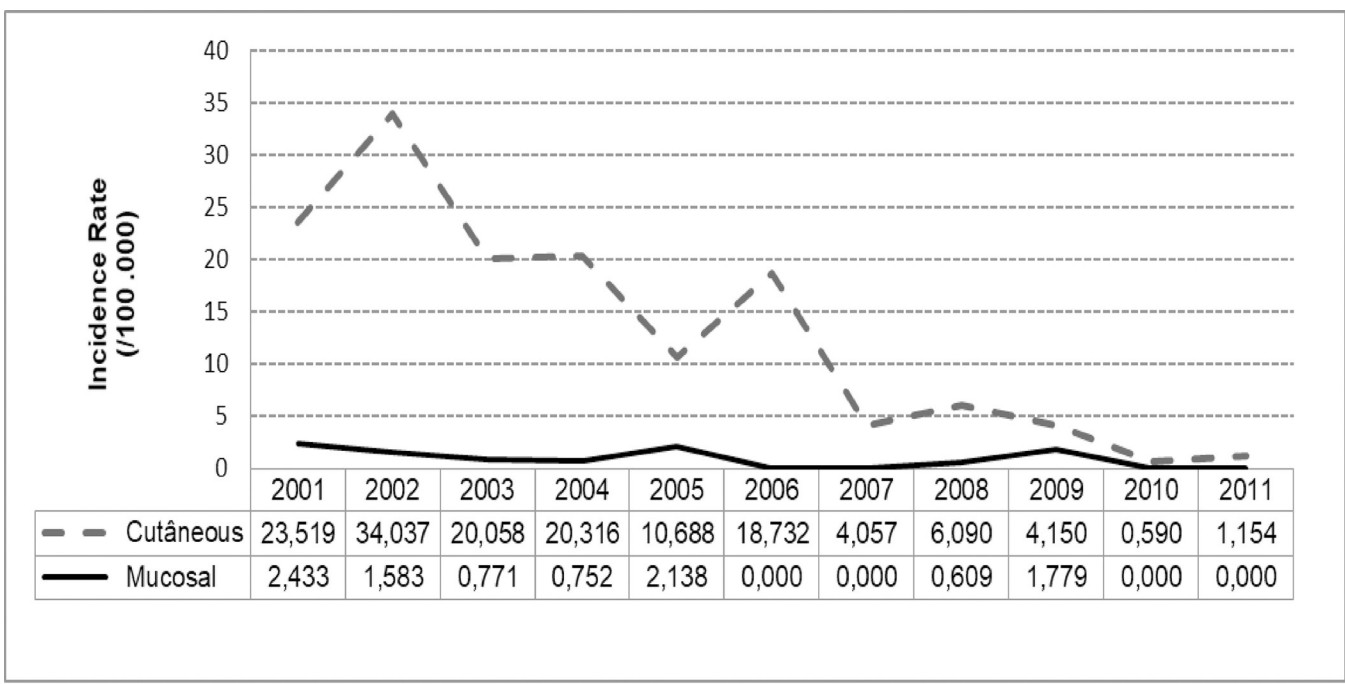

**Fig 8. Temporal evolution of the incidence rates according to tegumentary leishmaniasis clinical manifestations in the municipality of Angra dos Reis, Brazil, from 2001 to 2011.** FIOCRUZ, 2017.

The values predicted for CL incidence rate by 100,000 inhabitants, based on models A and C are in Fig 9.

## Discussion

Based on this study, we observed that the state of Rio de Janeiro and the two municipalities evaluated in it presented a higher incidence of CL and ML up to 2006, with a decline from 2007 on. It was also possible to observe that there was a temporal dependence on the occurrence of cases of the two clinical forms (CL and ML) in the municipalities of Angra dos Reis and Rio de Janeiro, and in the state of Rio de Janeiro. This study showed that the annual CL and ML incidence tends to increase.

The literature shows that the ATL distribution with CL incidence increase, followed by subsequent periods without notifications of this clinical form, as observed in some municipalities of the state of Rio de Janeiro, such as for example, the North and Northeast regions of the state, can be explained by the variation on the number of susceptible individuals in a location. In the state of Rio de Janeiro, from 2001 to 2013, ML represented 9.07% of all ATL cases, with a ratio of 10 CL: 1 ML [11]. It was suggested that in areas with lower infection prevalence, such as the Brazilian Southeast Region, the proportion of the mucosal leishmaniasis is increased, indicating the influence of the level of endemicity in the form of the clinically manifested disease. The low proportion of the mucosal leishmaniasis in old endemic areas and with high infection prevalence could be the consequence of a better adaptation between host, parasite and vector [11]. The low susceptibility to the disease in a community or group (cases treated and healed, spontaneous healing, sub-clinical infections) occurs due to the acquisition of long lasting resistance of previously infected individuals [1]. Thus, only when a subsequent accumulation of susceptible individuals occurs, by the birth of new individuals, a threshold enough to start an epidemic is reached. Migration may also be responsible for the variation of the susceptible pool of

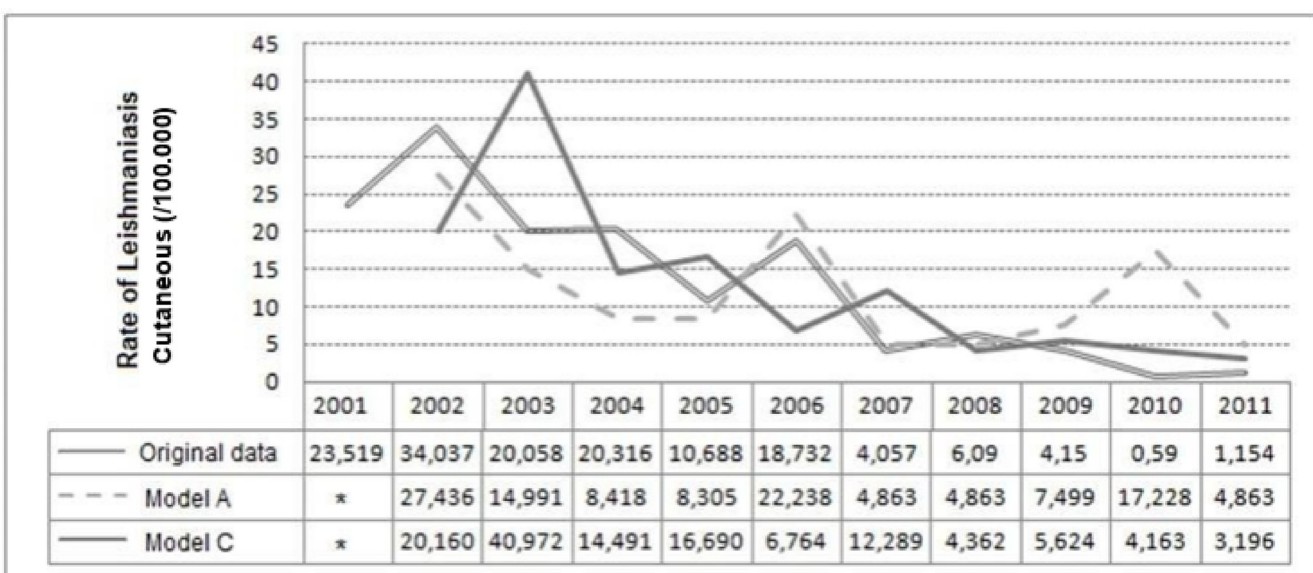

* Predicted values of rates not calculated due to the dependency of mucosal leishmaniasis incidencerate of the previous years

**Fig 9. Prediction of cutaneous leishmaniasis rates (/100,000) in the municipality of Angra dos Reis, Brazil, from 2001 to 2011.** FIOCRUZ, 2017. Model A: Cutaneous rate = Mucosal rate of the previous year, Model C: Cutaneous rate = Mucosal rate + Cutaneous rate of the previous year.

individuals, either by the immigrants getting in touch with the parasite of the region into which they displace, or because the infected immigrants carry a parasite into an area in which individuals are susceptible. Particularly, new cases of human CL can be secondary to the introduction of infected people or domestic animals, sick or not, into a new geographic vulnerable area (presence of vectors and susceptible persons). This introduces different aspects on the disease epidemiology and on leishmaniasis clinical severity (ecologic characteristics, circulating *Leishmania* spp., and population characteristics) including the risk of ML [10,24].

Although the cutaneous rate increases concomitantly with the annual mucosal rate, when controlled by the previous cutaneous rate in the state of Rio de Janeiro, a singular behavior is observed in the municipality of Rio de Janeiro. The annual cutaneous rate depends on ML incidence rate of the two previous years. High ML values in a year result on the reduction of CL rates two years later, regardless the CL values of the previous year. The municipality of Rio de Janeiro displays a differentiated situation regarding other municipalities of the state of. There are two forests in Rio de Janeiro city considered as potential big reservoirs of parasites and vectors, phlebotomine vectors are not frequently reported in the surroundings. These forests are the Tijuca Forest, the greatest tropical forest in an urban area in the world, and the Pedra Branca State Park [20]. However, CL and ML cases, as well as the presence of phlebotomine vectors are not reported in the surroundings of the Tijuca Massif, where the original arboreal vegetation was almost entirely substituted by reforesting with exotic species [20]. CL and ML cases are only originated in peri-urban areas of the Pedra Branca State Park, where the primary vegetation still survives. In 1922 the first epidemic in the Tijuca Massif was registered in Rio de Janeiro municipality. After fifty years without new cases, in 1973 there was an ATL epidemic in the slopes of the Pedra Branca Massif, in the Western Region of the municipality of Rio de Janeiro, a place where ATL transmission is still active, with occurrence of sporadic cases and outbreaks [25]. To date, no wild transmission cycle involving *L. (V.) braziliensis* was described in those areas, only the presence of infected domestic dogs, horses and cats, associated with the presence

of Phlebotominae adapted to the modified environments [9,10,20]. The demographic increase, associated with a migratory flow from the interior of the state of Rio de Janeiro and from other Brazilian states [26], favored a new urbanization dynamics. It allowed the entry of susceptible population groups or those with ATL into modified environments, already with the presence of vectors, favoring and widening the transmission of the endemics in those areas, without the necessity of the existence of a specific reservoir [10,14,27].

Antimoniate (MA) (Glucantime™) has been the most used ATL treatment, particularly in Brazil [1]. World Health Organization [2] recommends ATL treatment with 20 mg $Sb^{5+}$/kg/ day intramuscular (IM) during 20 days for CL and during 30 days for ML with no limits for maximum daily dose. Brazilian Ministry of Health recommends the same dose as but limited to a maximum daily dose of 1,215 mg $Sb^{5+}$, which is equivalent to 3 MA vials or 15 mL [1]. An alternative treatment plan for ATL is the use of 5 mg $Sb^{5+}$/kg/day IM during 30 days for CL, and until mucosal lesion healing, with a maximum of 120 doses for ML [28].

In the period from 2001 to 2013, INI/FIOCRUZ was responsible for the treatment of 27.4% of all CL cases, and of 76.2% of ML cases [18]. As INI/FIOCRUZ is a reference center for ATL treatment in the state of Rio de Janeiro, with one of the few otorhinolaryngology services in the country specialized in ML diagnosis and treatment, there may be cases wrongly notified as autochthonous of Rio de Janeiro municipality; many people come from other municipalities or states to receive treatment and state as their addresses those of their relatives living in the municipality [18]. Additionally, considering that a period of several months or years may elapse between infection and the development of a mucosal lesion, the address of the patient at the time of the diagnosis may be different from the area where he really acquired the infection [29]. Data from INI/FIOCRUZ show that around 15% of treated patients report other Brazilian states as a possible area of infection. This is a limitation of the study, as we cannot get around this using the model used. The Brazilian health system could improve this data collection burden through integration between the health information systems of different states and municipalities. This can be done through digital platforms that allow the exchange of information in real time, investing in the training of health professionals to collect epidemiological data and continuous training can help minimize errors in identifying the origin of the infection [15,18,30,31]. It is thus possible that those notified cases have different characteristics regarding the parasite or human host usually found in the state of Rio de Janeiro. Therefore, the failure in identifying the probable municipality of infection in the notification form can hamper the study of the temporal association between CL and ML cases, in both the municipality and the state of Rio de Janeiro. In other municipalities, such as Angra dos Reis, the temporal association is more evident. Even though, it was possible to observe a temporal association with concomitant increase of the incidence of both clinical manifestations in the same year, in the three investigated regions.

On the other hand, the concomitant increase of CL and ML annual incidence rates observed in the state of Rio de Janeiro, as well as in other localities, is in agreement with the higher incidence of concomitant ML in the state of Rio de Janeiro, although late ML is reported as the most frequent form in Brazil [15]. ML physio- pathogenesis is still not well-known [32]. It has not been established whether the time elapsed between CL and ML manifestations is the consequence of the persistence of the parasite in the host with subsequent triggering of the mucosal disease, or if it is possible that incipient mucosal lesions, not early identified or treated during CL active phase, may evolve to clinically evident mucosal lesions considered then as late [32]. The difficulty of patients to access health care units or their lack of financial resources to go to treatment referral centers to receive appropriate ATL monitoring can be associated with the development of ML. On the other hand, the long time elapsed between the beginning of the symptoms and the diagnosis of mucosal manifestation of ATL may reflect the

limited training of the physicians for the early diagnosis of ML [15,18]. The general medical practitioners are not always trained to adequately approach nasal complaints and the ear, nose and throat specialists frequently limit their intervention to conducting a nasal biopsy, which in the case of ML may be inconclusive [16]. The late diagnosis could also be partially explained by the delay of the patient in looking for medical care. However, since chronic nasal obstruction is a complaint that directly interferes in the quality of life and labor capacity of the individual, it is not probable that the delay occurs because of late search for treatment. In fact, ML patients usually report previous treatment for chronic rhinitis for long periods of time, without a defined etiological diagnosis [33]. At INI/FIOCRUZ, all ATL patients, regardless of mucosal complaints, are systematically examined by endoscopic methods, which allow early diagnosis and treatment of the mucosal lesions. The better diagnosis conditions and thorough search of early mucous membranes lesions can explain the higher incidence of concomitant ML in patients with incipient mucosal lesions and wrongly forwarded as having CL. Additionally, the cases of late ML treated at INI/FIOCRUZ had not usually been diagnosed and treated at the time of active CL in this institution, and therefore were not submitted to endoscopic examination of the mucous membranes in that occasion (not published data). It is possible, therefore, that those metastatic mucosal lesions were already present at the time of active CL, and that the adoption of routine endoscopic examination of the mucous membranes causes a decrease of the incidence of the late ML form, which, slowly evolving during several years, ends by leading to deformities, often irreversible. However, the reason why individuals infected with *Leishmania* can present mucosal involvement is not fully known [16]. Although the main agent involved in ML is *L. (V.) braziliensis*, other species have already been described, usually because of the proximity of the cutaneous lesion to mucous membranes: *L. (L.) amazonensis*, *L. (V.) guyanensis* and *L. (V.) panamensis* [1,24,26]. The association of the infection by *L. (V.) braziliensis* with this form of the disease suggests that, in addition to the host, other factors related to the parasite are relevant for the development of the mucosal disease. It is possible that leishmaniasis clinical manifestation depends on factors inherent to the parasite, on the natural resistance of the host and on the magnitude of the immunological response [16]. Studies indicate that the population of *L. (V.) braziliensis* circulating in the state of Rio de Janeiro presents an homogenous genetic pattern with little variability, and there seems to be no association between *Leishmania* genotipic patterns and ATL clinical manifestations [31,34,35]. However, different ATL clinical manifestations were previously associated with *Leishmania (V.) braziliensis* subpopulations, distributed among different locations in the Brazilian state of Bahia [29]. It is possible, therefore, that different *Leishmania* species and subpopulations present different temporal associations between CL and ML; to estimate the true ratio between CL and ML, long-term prospective studies with systematic endoscopic mucous membranes evaluation would be necessary even in asymptomatic patients.

Concluding, a temporal association between CL and ML cases was observed, with a concomitant increase of the incidence of both clinical forms in most of the localities of the state of Rio de Janeiro. This temporal association observed in the state raises the hypothesis that: either the mucosal lesions were already incipient from the beginning of CL manifestation, or the *Leishmania* species circulating in the state of Rio de Janeiro is able to produce early mucosal lesions.

## Supporting information

**S1 File.**
(XLSX)

**S2 File.**
(ZIP)

## Author Contributions

**Conceptualization:** Lucia Regina do Nascimento Brahim Paes, Cláudia Maria Valete-Rosalino.

**Data curation:** Lucia Regina do Nascimento Brahim Paes, Maria Inês Fernandes Pimentel, Cristina Maria Giodarno Dias.

**Formal analysis:** Lucia Regina do Nascimento Brahim Paes, Monica de Avelar F. M. Magalhães, Maria Inês Fernandes Pimentel, Luiz Eduardo Carvalho-Paes, Armando de Oliveira Schubach, Cláudia Maria Valete-Rosalino.

**Funding acquisition:** Lucia Regina do Nascimento Brahim Paes.

**Investigation:** Lucia Regina do Nascimento Brahim Paes, Monica de Avelar F. M. Magalhães.

**Methodology:** Lucia Regina do Nascimento Brahim Paes, Raquel de Vasconcellos Carvalhaes de Oliveira, Monica de Avelar F. M. Magalhães, Marcelo Rosandiski Lyra, Luiz Eduardo Carvalho-Paes, Ananda Dutra da Costa, Cristina Maria Giodarno Dias, Anísia Darc do Nascimento Brahim, Bruno Moreira de Carvalho, Ester Cleisla dos Anjos Soares, Armando de Oliveira Schubach, Cláudia Maria Valete-Rosalino.

**Project administration:** Lucia Regina do Nascimento Brahim Paes, Cláudia Maria Valete-Rosalino.

**Resources:** Lucia Regina do Nascimento Brahim Paes.

**Software:** Lucia Regina do Nascimento Brahim Paes, Monica de Avelar F. M. Magalhães.

**Supervision:** Lucia Regina do Nascimento Brahim Paes, Raquel de Vasconcellos Carvalhaes de Oliveira, Monica de Avelar F. M. Magalhães, Luiz Eduardo Carvalho-Paes, Bruno Moreira de Carvalho, Mauro Celio de Almeida Marzochi, Armando de Oliveira Schubach, Cláudia Maria Valete-Rosalino.

**Validation:** Lucia Regina do Nascimento Brahim Paes, Raquel de Vasconcellos Carvalhaes de Oliveira, Monica de Avelar F. M. Magalhães, Maria Inês Fernandes Pimentel, Ester Cleisla dos Anjos Soares.

**Visualization:** Lucia Regina do Nascimento Brahim Paes, Claudia Cristina Jardim Duarte, Ester Cleisla dos Anjos Soares.

**Writing – original draft:** Lucia Regina do Nascimento Brahim Paes, Ananda Dutra da Costa, Anísia Darc do Nascimento Brahim, Bruno Moreira de Carvalho, Claudia Cristina Jardim Duarte, Ester Cleisla dos Anjos Soares, Armando de Oliveira Schubach, Cláudia Maria Valete-Rosalino.

**Writing – review & editing:** Lucia Regina do Nascimento Brahim Paes, Monica de Avelar F. M. Magalhães, Maria Inês Fernandes Pimentel, Marcelo Rosandiski Lyra, Luiz Eduardo Carvalho-Paes, Ananda Dutra da Costa, Anísia Darc do Nascimento Brahim, Claudia Cristina Jardim Duarte, Mauro Celio de Almeida Marzochi, Ester Cleisla dos Anjos Soares, Armando de Oliveira Schubach, Cláudia Maria Valete-Rosalino.

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
