## [Decision Letter · Decision Letter 0]

15 Mar 2024

PONE-D-24-02709Comparison of the spatial and temporal distribution of cutaneous and mucosal leishmaniasis in the state of Rio de Janeiro between 2001 and 2011.PLOS ONE

Dear Dr. Valete-Rosalino,

Thank you for submitting your manuscript to PLOS ONE. After careful consideration, we feel that it has merit but does not fully meet PLOS ONE’s publication criteria as it currently stands. Therefore, we invite you to submit a revised version of the manuscript that addresses the points raised during the review process.

We look forward to receiving your revised manuscript.

Kind regards,

Mariana Lourenço Freire, Ph.D

Academic Editor

PLOS ONE

“This work was partially funded by Carlos Chagas Filho Foundation for Research Support of Rio de Janeiro State (FAPERJ) and the National Council for Research and Technological Development (CNPq), Brazil. We thank Marcia Andreia Sales da Costa, Luiz Torres Homem (INI/FIOCRUZ) and Marcelo Alves Coelho Junior (INI/FIOCRUZ) for support in graphic design.”

5. Please update your submission to use the PLOS LaTeX template. The template and more information on our requirements for LaTeX submissions can be found at http://journals.plos.org/plosone/s/latex.

6. Please amend your authorship list in your manuscript file to include authors Cláudia Maria Valete-Rosalino, Lucia Regina Brahim, Raquel de Vasconcellos Carvalhaes de Oliveira, Monica de Avelar F.M. Magalhães, Maria Inês Fernandes Pimentel, Marcelo Rosandiski Lyra, Luiz Eduardo Carvalho-Paes, Ananda Dutra da Costa, Cristina Maria Giodarno Dias, Anísia Darc do Nascimento Brahim, Bruno Moreira de Carvalho, Mauro Celio de Almeida Marzochi, Ester Cleisla dos Anjos Soares, and Armando de Oliveira Schubach.

7. Please amend your list of authors on the manuscript to ensure that each author is linked to an affiliation. Authors’ affiliations should reflect the institution where the work was done (if authors moved subsequently, you can also list the new affiliation stating “current affiliation:….” as necessary).

8. We note that Figures 1, 2 and 3 in your submission contain [map/satellite] images which may be copyrighted. All PLOS content is published under the Creative Commons Attribution License (CC BY 4.0), which means that the manuscript, images, and Supporting Information files will be freely available online, and any third party is permitted to access, download, copy, distribute, and use these materials in any way, even commercially, with proper attribution. For these reasons, we cannot publish previously copyrighted maps or satellite images created using proprietary data, such as Google software (Google Maps, Street View, and Earth). For more information, see our copyright guidelines: http://journals.plos.org/plosone/s/licenses-and-copyright.

1. You may seek permission from the original copyright holder of Figures 1, 2 and 3 to publish the content specifically under the CC BY 4.0 license. 

Reviewers' comments:

Reviewer's Responses to Questions

**Comments to the Author**

1. Is the manuscript technically sound, and do the data support the conclusions?

Reviewer #1: Yes

Reviewer #2: Partly

Reviewer #3: No

2. Has the statistical analysis been performed appropriately and rigorously? 

Reviewer #1: Yes

Reviewer #2: I Don't Know

Reviewer #3: I Don't Know

3. Have the authors made all data underlying the findings in their manuscript fully available?

Reviewer #1: Yes

Reviewer #2: Yes

Reviewer #3: Yes

4. Is the manuscript presented in an intelligible fashion and written in standard English?

Reviewer #1: Yes

Reviewer #2: Yes

Reviewer #3: Yes

5. Review Comments to the Author

Reviewer #1: Dear authors,

The manuscript is easy to read and understand.

It is clear, the aims are fulfilled by the methodology, the results are well described and the discussion is concise.

Overall the manuscript is very good and I strongly recommend it with minor revision.

Minor revisions:

1. I missed an explanation of why the authors used data on cutaneous and mucosal leishmaniasis between 2001 and 2011 since we are in 2024.

2. I strongly recommend updating references with more recent citations. The newest publication cited is from 2017 (7 years old).

Reviewer #2: The manuscript addresses a regional health issue of interest. It utilizes consistent information from national epidemiological data spanning a wide temporal spectrum (2001-2011). This positions the article as one of interest for publication. However, for a more comprehensive spatial conclusion, spatial and geographical data could have been included to better address the hypotheses.

Here are queries, concerns, and recommendations regarding the work conducted:

The authors choose two municipalities (RJ and Angra do Reis) to evaluate and describe in the general objective the need to study the temporal and spatial distribution of cases in these municipalities. However, it is only in the results that the rationale for their choice of municipalities becomes clear. It is recommended to incorporate more information about these municipalities in the introduction, providing context for their selection.

Figure 1: The municipalities under study are not clearly identifiable. The figure states "city of Angra do Reis". Is it possible that instead of "city," it should say "municipality"?

Statistics: It is proposed to describe in more detail each of the variables used and their meanings. Additionally, it is also necessary to declare how the temporal variables were chosen and their times for each model (1 year, 2 years, why?).

Figure 4: While it is very useful to have all municipalities displayed simultaneously for each year, it is very difficult due to its size to visualize the increase or decrease in the incidence rate of cases of the municipalities of interest (which are described in the objective). It is recommended to highlight the municipalities under study.

In the discussion, the authors declare the difficulty of recognizing the origin of the municipality where the patient was infected. "Data from INI/FIOCRUZ show that approximately 15% of treated patients report, as a possible infection area, other Brazilian states." In this regard, I inquire, how do you consider incorporating this bias into the model? How were the 15% of individuals who reported that their municipality of residence and infection were different incorporated into the model? Lastly, how do you think the Brazilian healthcare system could improve this data collection burden? I suggest that these responses be added to the discussion.

Reviewer #3: Overall, at the objective is not clear what is the importance in compare the cases of whole state with specific municipalities, as Rio de Janeiro and Angra dos Reis, and also with the State. The methodology and presentation of model results are very confused, which compromises the interpretation of the study, as well as its purpose.

The introduction section is long and wordy.

Please, add line numbers.

Specific comments are bellow:

1. Regarding the sentences related to the reference 1:

“In the state of Rio de Janeiro, ATL is almost exclusively caused by this Leishmania species and it occurs in areas where phlebotomine vectors of peridomestic habits, such as Lutzomyia intermedia and Lutzomyia migonei, predominate.”

“Frequently, the infection occurs in domestic dogs, horses and eventually in domestic cats.”

Please cite the author(s) who found these informations. Although the Brazilian Ministry of Healt document is citing that data, there was not this Institution that did these studies and found these observations.

Please, cite references that found domestic dogs, horses and cats infected by the parasites. In the same way the authors who observed the predominant vectors species in Rio de Janeiro State. And please, pay attention in other citations regarding this reference “1”.

2. Regarding this phrase: “To our knowledge, only one autochthonous case of L. (Leishmania) amazonensis and two unusual cases of cutaneous leishmaniasis (CL) caused by L. (L.) infantum were described in the state of Rio de Janeiro.”

Please, cite the data up to the period in which this information is known.

3. About: “The long and variable incubation period could also hamper the epidemiological analyses of correlation between the occurrence or the introduction of new cases and the possible outbreak of secondary cases of CL and ML”

About “The long and variable incubation period”. In the previous phrase is citing an incubation period from 2 weeks to 2 months. Is this period long sufficiently to hamper the epidemiological analyses cited? Or are you referring to the period (12 years) of being infected after treatment, that is different of incubation period). Please, explain it better.

4. About the 4th paragraph. It is not recommended to start phrases with acronym. The introduction section is too large, thus please evaluate whether describing the immunology of the disease is important data for your study. Just in case, cut some issues to reduce the introduction section. It seems

5. “… in the municipality of Rio de Janeiro, is responsible for the treatment of most ML patients notified in the RJ state.”

Define RJ acronym.

6. “Studies in that institution have revealed that, in the state of Rio de Janeiro, ….”

Please replace “…in the state of Rio de Janeiro” for in that State, because Rio de Janeiro is repeating close.

7. “…2005 (351 cases), and after that year there was a reduction of 85.07% in ATL incidence rate, decreasing from 1.44/100.000 inhabitants in 2004 to 0.20/100.000 inhabitants in 2013.”

Please explain it better. The year after 2005 is 2006, rather than 2004… This phrase is confused. In fact, the whole paragraph is confused, I suggest you resume the ideas, also because the introduction section is large.

8. “To date, there is no study on the spatial and temporal distribution of CL and ML cases. The objective of the present study was to compare the spatial and temporal distribution of CL cases with ML cases in the state of Rio de Janeiro and the municipalities of Rio de Janeiro and Angra dos Reis, between 2001 and 2011.”

This sentence is long and repeating terms. I suggest you clean up the text.

9. “160 neighborhoods, grouped into 33 administrative regions and seven district councils”.

“… the minimum absolute temperature registered in Angra dos Reis, since 1961, was 9.4°C on August 12, 1988, and the maximum absolute temperature was 39.3°C on February, 12, 1966.”

Look that these periods were a long time ago. Please evaluate if these informations are important for your study. It seems some data not relevant for your study and so the text is very long.

How many municipalities are in the state. This information is important, once they are focused on the analyzes. Are they 33? Please, specific it.

10. Cite “(Figure 1)” after the 1st phrase of second paragraph of the methodology section, rather than at the end of that paragraph.

11. Fig 1

Legend of Figure: substitute “regions” for “Region”.

Brazil map: Please, add the limit of other Brazilian Regions, it seems that there are only the Region of Rio de Janeiro and one other….

Please add the limit of other states, at least in the highlighted Region, where Rio de Janeiro state is located. It seems that there are only the State of Rio de Janeiro and one other.

Down map: Please add grid of geographical coordinates .

At the map is referring the City of Angra dos Reis and Rio de Janeiro and in the Legend of the Figure they are as Municipality. So, please, define if they are City or Municipality, because both terms mean different issues.

12. Population and ATL section

“… de Janeiro. (SESRJ), considering”. Here are point and comma.

Please, explain that you are calculating the incidence, because is not clear in this phrase why the “population” is considering here. In addition, explain how about the cases, their Municipalities are defined by Probably local of infection? Residence? Notification?

Are their years referring to the year of 1st symptoms or of notification?

13. “… the spatial and temporal analysis of CL and ML occurrence”.

please mention the name of the spatial and temporal analysis here to inform the reader of the next steps to be explained. In addition, cite for what do you use the ArcGis software. Was it used to spatial and temporal analysis?

14. “All analyses were conducted for the state of Rio de Janeiro and for the municipalities of Rio de Janeiro and Angra dos Reis”.

How are you comparing data of state of Rio de Janeiro and for the municipalities of Rio de Janeiro and Angra dos Reis in the analyzes? Please explain it better.

If the State have 33 municipalities, are you comparing two municipalities with the other 31?

15. “A negative binomial regression model was used to assess the temporal dependency between the series.”

What are you meaning with “series”?

16. Model C: Regarding “rate for the year of study”. What are you meaning with “the year of the study”? Perhaps you can represent the years with t and t-1 or t-2, when t=year… Could you please add the design of the models to be clearer? Perhaps using equations…

17. What are the independent variables of the models?

18. The dependent variables are the incidence per municipalities? For example: in model B the dependent variable is the incidence of MLt-2?

Without this both data and looking the results of models is difficult to understand the ideas of these analyzes.

I suggest you add in the results Section the table of model results, rather than the values in the body of the text.

19. “The model with smaller Akaike information criterion (AIC) was considered the best model.”

Is this considering for the choose of one of four models, or the choose of independent variables of best model for each one of four models?

20. “It is possible to observe that ML cases occurred in the same year or up to 5.5 years after the CL cases, with median of 1 year between both clinical forms.”

Please, explain better this observation, because it is not clear by just looking the Figures 2 and 3.

What are you meaning with “5.5 years”, once the figures are represented by complete years and not half a year.

Please add in the legends of Figures 2 and 3 that the incidence is per 100,000 habitants.

21. “Model C showed that ML rate (β1=6.9927, p=0.0231) tends to increase concomitantly with CL rate of the same year, even when controlled by the CL rate of the previous year (β2=0.3052, p=0.3039).”

This phrase is not clear, because the part “even when controlled by the CL rate of the previous year” had p value less than 0.05, that is not significant. Perhaps showing the table of model results (as suggested above) can elucidate the idea.

22. “Models A and C had good fit according to the deviance function… ”

I suggest you compare both models with statistical analysis, rather than just observing their values of p and AIC, as is in the text.

23. “In most localities the annual incidence rates of CL and ML increase concomitantly even when adjusted by the values of the incidence rates of CL of the previous year.”

Please, I suggest you specify the magnitude of this statement, just like "most" is how many percent (representing x% (x/x))? This can be mentioned in the result section and then make the statement more tangible.

24. “ATL distribution with CL incidence increase, followed by subsequent periods without notifications of this clinical form, as observed in some municipalities of the state of Rio de Janeiro, such as for example, the North and Northeast regions of the state, can be explained by the variation on the number of susceptible individuals in a location.”

How do you know the “variation on the number of susceptible individuals in a location”?

25. “In the state of Rio de Janeiro, from 2001 to 2013, ML represented 9.07% of all ATL cases, with a ratio of 10 CL: 1 ML.”

Is this 9.07% a percentage of all cases in Brazil in this period? Please explain it and add the reference. I also suggest you add the absolute numbers, such as 9.07% (x/x).

26. “The low susceptibility to the disease in a community or group (cases treated and healed, spontaneous healing, sub-clinical infections) occurs due to the acquisition of long lasting resistance of previously infected individuals.”

Please add reference, also for the follow phrase.

In addition, please, check if your examples of “low susceptible” are referring to “resistant population”, due to their immune development. When the population is compartmentalized according to dynamic disease (as demonstrated in the follow phrase of the text), the epidemiological concept is different for both susceptible and resistant individuals. The “low susceptible” individuals can be due to low number of them or individual characteristics, such as genetical immunity, sex, gender, and others….

27. “… where ATL would find an adequate and favorable environment for its development…”

Is the “disease ATL” that find adequate for its development, or is the dynamic cycle that could be maintained? It seems that the epidemiological terms are confused.

28. “…phlebotomine vectors are not reported in the surroundings of the Tijuca Massif, where the original arboreal vegetation was almost entirely substituted by reforesting with exotic species…”

Please add reference.

29. “The municipality of Rio de Janeiro displays a differentiated situation regarding other municipalities of the state of Rio de Janeiro.”

Please take of the second “Rio de Janeiro”. They are redundant.

30. “To date, no wild transmission cycle involving L. (V.) braziliensis was described in those areas, only the presence of infected domestic dogs, horses and cats, associated with the presence of Phlebotominae adapted to the modified environments.”

Please add a reference.

31. (Brito, 2016)

Please fit this reference according to PlosOne style, and also add it in the References section.

32. “It allowed the entry of susceptible population groups or those with ATL into modified environments, already with a high vector density, favoring and widening the transmission of the endemics in those areas, without the necessity of the existence of a specific reservoir.”

Please add the reference about “high density vector” at that area

33. “In the period from 2001 to 2013, INI/FIOCRUZ was responsible for the treatment of 27.4% of all CL cases, and of 76.2% of ML cases.”

Is this percentages regarding to the whole State or to the Country?

34. “the long time elapsed between the beginning of the symptoms and the diagnosis of mucosal manifestation of ATL”

What is this “long time”? Please, add a reference.

35. “Although the main agent involved in ML is L. (V.) braziliensis, other species have already been described, usually because of the proximity of the cutaneous lesion to mucous membranes: L. (L.) amazonensis, L. (V.) guyanensis and L. (V.) panamensis.”

Please, check if this reference is 25.

6. PLOS authors have the option to publish the peer review history of their article (what does this mean?). If published, this will include your full peer review and any attached files.

Reviewer #1: No

Reviewer #2: No

Reviewer #3: No

---

## [Author Response · Author response to Decision Letter 0]

22 Jun 2024

Dear Editor,

We appreciate the meticulous review process undertaken by the editorial

board and the valuable insights provided by the reviewers. The feedback has

been invaluable in improving the quality and accuracy of our work. We are

genuinely grateful for the time and attention dedicated to evaluating our

manuscript. Therefore, we are submitting a revised version of the manuscript

addressing all points raised during the review process.

1. Please ensure that your manuscript meets PLOS ONE's style

requirements, including those for file naming. DONE

2. Please note that PLOS ONE has specific guidelines on code sharing for

submissions in which author-generated code underpins the findings in

the manuscript. In these cases, all author-generated code must be made

available without restrictions upon publication of the work. Please review

our guidelines at https://journals.plos.org/plosone/s/materials-and-

software-sharing#loc-sharing-code and ensure that your code is shared

in a way that follows best practice and facilitates reproducibility and

reuse. DONE

3. Thank you for stating the following in the Acknowledgments

Section of your manuscript: “This work was partially funded by Carlos

Chagas Filho Foundation for Research Support of Rio de Janeiro State

(FAPERJ) and the National Council for Research and Technological

Development (CNPq), Brazil. We thank Marcia Andreia Sales da Costa,

Luiz Torres Homem (INI/FIOCRUZ) and Marcelo Alves Coelho Junior

(INI/FIOCRUZ) for support in graphic design.” We note that you have

provided funding information that is not currently declared in your

Funding Statement. However, funding information should not appear in

the Acknowledgments section or other areas of your manuscript. We will

only publish funding information present in the Funding Statement

section of the online submission form. Please remove any funding-

related text from the manuscript and let us know how you would like to

update your Funding Statement. Currently, your Funding Statement

reads as follows: “The author(s) received no specific funding for this

work.” Please include your amended statements within your cover letter;

we will change the online submission form on your behalf.

In our submission, we would like to clarify that no specific funding was

received for the research presented in this article. However, during her

postgraduate studies, one of the authors received a scholarship that

significantly supported her studies and contributed to the completion of

the research. For this reason, we had thanked the funding agency

responsible for granting the scholarship in the acknowledgments section

of our manuscript. The people mentioned were the IT technicians from INI-

FIOCRUZ who provide us with support on several occasions, whether with

the network or machine problems.

Acknowledgements Line 661

4. We note that your Data Availability Statement is currently as follows: [All

relevant data are within the manuscript and its Supporting Information

files.]

Please confirm at this time whether or not your submission contains all raw data

required to replicate the results of your study. Authors must share the “minimal

data set” for their submission. PLOS defines the minimal data set to consist of

the data required to replicate all study findings reported in the article, as well as

related metadata and methods (https://journals.plos.org/plosone/s/data-

availability#loc-minimal-data-set-definition). For example, authors should submit

the following data: - The values behind the means, standard deviations and

other measures reported; - The values used to build graphs; - The points

extracted from images for analysis. Authors do not need to submit their entire

data set if only a portion of the data was used in the reported study. If your

submission does not contain these data, please either upload them as

Supporting Information files or deposit them to a stable, public repository and

provide us with the relevant URLs, DOIs, or accession numbers. For a list of

recommended repositories, please see

https://journals.plos.org/plosone/s/recommended-repositories. If there are

ethical or legal restrictions on sharing a de-identified data set, please explain

them in detail (e.g., data contain potentially sensitive information, data are

owned by a third-party organization, etc.) and who has imposed them (e.g., an

ethics committee). Please also provide contact information for a data access

committee, ethics committee, or other institutional body to which data requests

may be sent. If data are owned by a third party, please indicate how others may

request data access.

Yes, we confirm that all the important data necessary to replicate the

results of our study are included in the article and its Supplementary

Information files. This data is unrestricted and publicly accessible and can

be accessed through the following links:

https://censo2010.ibge.gov.br/;

https://portalsinan.saude.gov.br/

https://portal.inmet.gov.br/

The incidence in figures 2 page 13 and 3 page 15 . Including in the

table 1page 16, figures 4 and 5 page 17, 6 page 16, 7 page 17 and figure

8 and 9 page 19.

5. Please update your submission to use the PLOS LaTeX template. The

template and more information on our requirements for LaTeX

submissions can be found at http://journals.plos.org/plosone/s/latex.

DONE

6. Please amend your authorship list in your manuscript file to include

authors Lucia Regina Brahim, Cláudia Maria Valete, Raquel de

Vasconcellos Carvalhaes de Oliveira, Monica de Avelar F.M. Magalhães,

Maria Inês Fernandes Pimentel, Marcelo Rosandiski Lyra, Luiz Eduardo

Carvalho-Paes, Ananda Dutra da Costa, Cristina Maria Giodarno Dias,

Anísia Darc do Nascimento Brahim, Bruno Moreira de Carvalho, Claudia

Cristina Jardim Duarte, Mauro Celio de Almeida Marzochi, Ester Cleisla

dos Anjos Soares, and Armando de Oliveira Schubach. DONE

The authors of work were duly included in the manuscript.

Page 1

7. Please amend your list of authors on the manuscript to ensure that each

author is linked to an affiliation. Authors’ affiliations should reflect the

institution where the work was done (if authors moved subsequently, you can

also list the new affiliation stating “current affiliation:….” as necessary).

DONE

The places of work were duly included in the manuscript. Page 1

8. We note that Figures 1, 2 and 3 in your submission contain

[map/satellite] images which may be copyrighted. All PLOS content is

published under the Creative Commons Attribution License (CC BY 4.0),

which means that the manuscript, images, and Supporting Information

files will be freely available online, and any third party is permitted to

access, download, copy, distribute, and use these materials in any way,

even commercially, with proper attribution. For these reasons, we cannot

publish previously copyrighted maps or satellite images created using

proprietary data, such as Google software (Google Maps, Street View,

and Earth). For more information, see our copyright guidelines:

http://journals.plos.org/plosone/s/licenses-and-copyright.

We require you to either (1) present written permission from the copyright

holder to publish these figures specifically under the CC BY 4.0 license, or (2)

remove the figures from your submission:

1. You may seek permission from the original copyright holder of Figures 1, 2

and 3 to publish the content specifically under the CC BY 4.0 license. We

recommend that you contact the original copyright holder with the Content

Permission Form (http://journals.plos.org/plosone/s/file?id=7c09/content-

permission-form.pdf) and the following text: “I request permission for the open-

access journal PLOS ONE to publish XXX under the Creative Commons

Attribution License (CCAL) CC BY 4.0

(http://creativecommons.org/licenses/by/4.0/). Please be aware that this license

allows unrestricted use and distribution, even commercially, by third parties.

Please reply and provide explicit written permission to publish XXX under a CC

BY license and complete the attached form.” Please upload the completed

Content Permission Form or other proof of granted permissions as an "Other"

file with your submission. In the figure caption of the copyrighted figure, please

include the following text: “Reprinted from [ref] under a CC BY license, with

permission from [name of publisher], original copyright [original copyright year].”

2. If you are unable to obtain permission from the original copyright holder to

publish these figures under the CC BY 4.0 license or if the copyright holder’s

requirements are incompatible with the CC BY 4.0 license, please either i)

remove the figure or ii) supply a replacement figure that complies with the CC

BY 4.0 license. Please check copyright information on all replacement figures

and update the figure caption with source information. If applicable, please

specify in the figure caption text when a figure is similar but not identical to the

original image and is therefore for illustrative purposes only. The following

resources for replacing copyrighted map figures may be helpful: USGS National

Map Viewer (public domain): http://viewer.nationalmap.gov/viewer/ The

Gateway to Astronaut Photography of Earth (public domain):

http://eol.jsc.nasa.gov/sseop/clickmap/ Maps at the CIA (public domain):

https://www.cia.gov/library/publications/the-world-factbook/index.html and

https://www.cia.gov/library/publications/cia-maps-publications/index.html NASA

Earth Observatory (public domain): http://earthobservatory.nasa.gov/ Landsat:

http://landsat.visibleearth.nasa.gov/ USGS EROS (Earth Resources

Observatory and Science (EROS) Center) (public domain):

http://eros.usgs.gov/# Natural Earth (public domain):

http://www.naturalearthdata.com/

We confirm that images 1, 2 and 3 presented in our submission were

created using the ARCHx program at ICICT - FIOCRUZ - Institute of

Scientific and Technological Communication and Information - Oswaldo

Cruz Foundation, they are copyright maps prepared with public data from

SINAN and IBGE by a from our authors specialized in geoprocessing,

therefore they were ours and generated according to our results.

Confirming then, that the figures do not come from any external location,

website or public place that requires authorization. Below are the links to

the database we use for the work.

https://censo2010.ibge.gov.br/

https://portalsinan.saude.gov.br/

https://portal.inmet.gov.br/

Reviewer #1: Dear authors,

The manuscript is easy to read and understand.

It is clear, the aims are fulfilled by the methodology, the results are well

described and the discussion is concise.

Overall the manuscript is very good and I strongly recommend it with minor

revision.

Minor revisions:

1. I missed an explanation of why the authors used data on cutaneous and

mucosal leishmaniasis between 2001 and 2011 since we are in 2024.

2. I strongly recommend updating references with more recent citations. The

newest publication cited is from 2017 (7 years old).

Regarding your first point, we apologize for any confusion regarding the

timeframe of our study. The reason we used data on cutaneous and

mucosal leishmaniasis from 2001 to 2011 is because this period

corresponds to the years when our database was complete with all the

variables used and all the patients included through the State Health

Secretariat of Rio de Janeiro (SESRJ) and the Notifiable Diseases

Information System (SINAN) of the Ministry of Health, along with the

population estimates of these municipalities through census data from

the Brazilian Institute of Geography and Statistics (IBGE). We have

clarified this point in the manuscript.

Regarding your second point and recognizing the importance of updating

references with more recent citations, we reviewed the literature and

included 6 new references.

Literature update.

Version with corrections.

Introduction: 1st paragraph lines 54, 59 and 67, second paragraph line 69

Materials and methods page 23 3rd paragraph Lines 571

Discussion page 25 6th paragraph lines 639, 642

Reviewer #2: The manuscript addresses a regional health issue of interest. It

utilizes consistent information from national epidemiological data spanning a

wide temporal spectrum (2001-2011). This positions the article as one of

interest for publication. However, for a more comprehensive spatial conclusion,

spatial and geographical data could have been included to better address the

hypotheses.

Here are queries, concerns, and recommendations regarding the work

conducted:

The authors choose two municipalities (RJ and Angra do Reis) to evaluate and

describe in the general objective the need to study the temporal and spatial

distribution of cases in these municipalities. However, it is only in the results

that the rationale for their choice of municipalities becomes clear. It is

recommended to incorporate more information about these municipalities in the

introduction, providing context for their selection. DONE

Although we carried out the analyzes in 92 municipalities in the State of Rio de

Janeiro, the municipalities of Angra dos Reis and Rio de Janeiro (capital of the

State) were chosen for this study, as they were the ones that presented the

most significant data in relation to LC and LM incidence rates from 2001 to 2011

Introduction page 5 lines 159 to 163.

Figure 1: The municipalities under study are not clearly identifiable. The figure

states "city of Angra do Reis". Is it possible that instead of "city," it should say

"municipality"?DONE

Statistics: It is proposed to describe in more detail each of the variables used

and their meanings. Additionally, it is also necessary to declare how the

temporal variables were chosen and their times for each model (1 year, 2 years,

why?). DONE

Materials and Methods - Statistical Analysis - Page 10 to 11 line 280 to 308

Figure 4: While it is very useful to have all municipalities displayed

simultaneously for each year, it is very difficult due to its size to visualize the

increase or decrease in the incidence rate of cases of the municipalities of

interest (which are described in the objective). It is recommended to highlight

the municipalities under study. DONE

Changes have been made.

Figure 2 page 11 and figure 3 page 13

In the discussion, the authors declare the difficulty of recognizing the origin of

the municipality where the patient was infected. "Data from INI/FIOCRUZ show

that approximately 15% of treated patients report, as a possible infection area,

other Brazilian states." In this regard, I inquire, how do you consider

incorporating this bias into the model? How were the 15% of individuals who

reported that their municipality of residence and infection were different

incorporated into the model? Lastly, how do you think the Brazilian healthcare

system could improve this data collection burden? I suggest that these

responses be added to the discussion. DONE

Discussion page 23 line 571 to 580

Data from INI/FIOCRUZ show that around 15% of treated patients report other

Brazilian states as a possible area of infection. This is a limitation of the study,

as we cannot get around this using the model used. The Brazilian health system

could improve this data collection burden through integration between the

health information systems of different states and municipalities. This can be

done through digital platforms that allow the exchange of information in real

time, investing in the training of health professionals to collect epidemiological

data and continuous training can help minimize errors in identifying the origin of

the infection.

Reviewer #3: Overall, at the objective is not clear what is the importance in

compare the cases of whole state with specific municipalities, as Rio de Janeiro

and Angra dos Reis, and also with the State. The methodology and

presentation of model results are very confused, which compromises the

interpretation of the study, as well as its purpose.

The introduction section is long and wordy.

Please, add line numbers.

Specific comments are bellow:

1. R

---

## [Decision Letter · Decision Letter 1]

22 Jul 2024

Comparison of the spatial and temporal distribution of cutaneous and mucosal leishmaniasis in the state of Rio de Janeiro between 2001 and 2011

PONE-D-24-02709R1

Dear Dr. Valete-Rosalino,

We’re pleased to inform you that your manuscript has been judged scientifically suitable for publication and will be formally accepted for publication once it meets all outstanding technical requirements.

Kind regards,

Mariana Lourenço Freire, Ph.D

Academic Editor

PLOS ONE

Additional Editor Comments (optional):

Reviewers' comments:

Reviewer's Responses to Questions

**Comments to the Author**

1. If the authors have adequately addressed your comments raised in a previous round of review and you feel that this manuscript is now acceptable for publication, you may indicate that here to bypass the “Comments to the Author” section, enter your conflict of interest statement in the “Confidential to Editor” section, and submit your "Accept" recommendation.

Reviewer #1: All comments have been addressed

Reviewer #2: All comments have been addressed

2. Is the manuscript technically sound, and do the data support the conclusions?

Reviewer #1: Yes

Reviewer #2: Partly

3. Has the statistical analysis been performed appropriately and rigorously? 

Reviewer #1: Yes

Reviewer #2: I Don't Know

4. Have the authors made all data underlying the findings in their manuscript fully available?

Reviewer #1: Yes

Reviewer #2: Yes

5. Is the manuscript presented in an intelligible fashion and written in standard English?

Reviewer #1: Yes

Reviewer #2: Yes

6. Review Comments to the Author

Reviewer #1: Dear authors, I appreciate your time and thank you for address all the comments and provide explanation.

Reviewer #2: Dear Authors, Thank you for addressing all the review comments and for the significant improvements made to the manuscript. Your clarifications and updates have effectively resolved the concerns raised. I believe the manuscript is now suitable for publication, as it continues to provide valuable insights into the public health issue of leishmaniasis in the region. Best regards,

7. PLOS authors have the option to publish the peer review history of their article (what does this mean?). If published, this will include your full peer review and any attached files.

Reviewer #1: No

Reviewer #2: No

---

## [Editor Report · Acceptance letter]

10 Oct 2024

PONE-D-24-02709R1 

PLOS ONE

Dear Dr. Valete-Rosalino, 

I'm pleased to inform you that your manuscript has been deemed suitable for publication in PLOS ONE. Congratulations! Your manuscript is now being handed over to our production team.

Kind regards, 

on behalf of

Dr. Mariana Lourenço Freire 

Academic Editor

PLOS ONE